# Development of Simultaneous Determination Method of Pesticide High Toxic Metabolite 3,4-Dichloroaniline and 3,5 Dichloroaniline in Chives Using HPLC-MS/MS

**DOI:** 10.3390/foods12152875

**Published:** 2023-07-28

**Authors:** Yibo Dong, Xiaolong Yao, Wanping Zhang, Xiaomao Wu

**Affiliations:** 1Institute of Crop Protection, Guizhou University, Guiyang 550025, China; 13511972016@163.com (Y.D.); xiaolongyao1021@163.com (X.Y.); 2Institute of Vegetable Research, Guizhou University, Guiyang 550025, China; 3Provincial Key Laboratory for Agricultural Pest Management in Mountainous Region, Guizhou University, Guiyang 550025, China

**Keywords:** 3,4-dichloroaniline, 3,5-dichloroaniline, simultaneous determination, UPLC-MS/MS, chive

## Abstract

3,4-dichloroaniline (3,4-DCA) and 3,5-dichloroaniline (3,5-DCA) are, respectively, the primary metabolites deriving from the breakdown of phenylurea herbicides and dicarboximide fungicides in both soils and plants, whose residues in vegetable products have a heightened concern considering their higher health risks to humans and greater toxicity than the parent compounds in the environment. In this study, a sensitive high-performance liquid chromatography-tandem mass spectrometry (HPLC-MS/MS) method was developed for the simultaneous determination of 3,4-DCA and 3,5-DCA residues in chive products based on the optimization of HPLC-MS/MS chromatographic and mass-spectrometric conditions using the standard substances and the modified QuEChERS preparation technique. The preparation efficiency of 3,4-DCA and 3,5-DCA from chive samples showed that acetonitrile was the best extractant. The combination of the purification agent graphite carbon black + primary secondary amine and the eluting agent acetonitrile + toluene (4:1, *v*/*v*) had a satisfactory purification effect. The linear correlation coefficients (*R*^2^) were more than 0.996 with the six concentration range of 0.001–1.000 mg/L for 3,4-DCA and 3,5-DCA. The limit of detection and limit of quantitation of this method was 0.6 and 2.0 µg/kg for 3,4-DCA, as well as 1.0 and 3.0 µg/kg for 3,5-DCA, respectively. The matrix effect range of 3,4-DCA and 3,5-DCA in chive tissues was from −9.0% to −2.6% and from −4.4% to 2.3%, respectively. The fortified recovery of 3,4-DCA and 3,5-DCA in chive samples at four spiked levels of 0.001–1.000 mg/kg was 75.3–86.0% and 78.2–98.1%, with the relative standard deviation of 2.1–8.5% and 1.4–11.9%, respectively. The limit of detection (*LOD*) and limit of quantification (*LOQ*) of the method were 0.6, 2.0, and 1.0, 3.03 for 4-DCA and 3,5-DCA, respectively. This study highlights that the analytical method established here can efficiently and sensitively detect residues of 3,4-DCA and 3,5-DCA residues for monitoring chive products. The method was successfully applied to 60 batches of actual vegetable samples from different regions.

## 1. Introduction

The 3,4-dichloroaniline (3,4-DCA) and 3,5-dichloroaniline (3,5-DCA) are the two most commonly utilized derivatives of the dichloroaniline family of aromatic amines [1,2]. In agricultural production, dichloroaniline is a primary metabolite of pesticides with high bioaccumulation levels and toxic health risks to humans [3,4]. These compounds thus become a type of pollutants of particular concern when they enter the environment. The primary physical and chemical properties of the two DCAs are shown in Appendix A. 

3,4-DCA is a precursor and intermediate in the chemical synthesis of a variety of phenylurea herbicides (e.g., diuron and linuron) and is also a primary metabolite of these pesticides in plants and the environment, with higher toxicity and longer persistence in the environment than parent compounds [5,6]. 3,4-DCA is a relatively mobile compound. Recent research has revealed that it spreads easily into the natural environment and is generally difficult to degrade [7,8]. Therefore, this compound is often detected in the environment, especially in water [9,10]. According to the 2012 United States Geological Survey report, 3,4-DCA was detected in the Thorpe and Chattacooqui rivers at concentrations as high as 68.2 ng/L. Toxicology studies have demonstrated that 3,4-DCA influences the development and reproduction of aquatic organisms and is more toxic than diuron [11,12,13]. 3,4-DCA could still lead to dose-dependent growth retardation and deformities in young *Gobiocypris rarus* [14]. In freshwater rotifers, 2.5 mg/L of 3,4-DCA could significantly decrease the generation time and reproductive rate [15]. Other studies also show that only 9 μg/L of 3,4-DCA could influence the reproduction rate of *Daphnia macroflora* [16]. 

3,5-DCA is a primary metabolite of dicarboximide fungicides such as procymidone, iprodione, and vinclozolin in plants and environments. Moreover, 3,5-DCA is the most toxic isomer of dichloroaniline due to its relatively high neurotoxicity [17]. Ambrus reported that procymidone iprodione and vinclozolin could be metabolized to 3,5-DCA in tomatoes, cucumbers, and other plants [18]. In addition, Rifai et al. [19] found that 3,5-DCA was also a photolysis product of procymidone. The United States Environmental Protection Agency (EPA) reported that 3,5-DCA deriving from procymidone and iprodione, was a carcinogenic product with a higher dietary risk than the parent compound. Thus, the EPA considered it to be a carcinogen [20]. Valentovic et al. showed that 3,5-DCA caused kidney damage and acute toxicity to the kidney and liver within 24 h [21]. Lai et al. still found that the acute toxicity of 3,5-DCA to rats was nearly ten times higher than that of its parent procymidone through computer simulation [22]. Meanwhile, Lai et al. found that both procymidone and iprodione could be metabolized into 3,5-DCA in zebrafish, and its toxicity to adult zebrafish was higher than that of parent compounds, and LC_50_ was 18–72% lower than that of parent compounds at 96 h [22].

Since both 3,4-DCA and 3,5-DCA have higher bioaccumulation levels and more toxic health risks in humans than their parent compounds [23,24], they may damage the endocrine system when entering into the human and animal bodies through the food chain. In the production of agricultural products (e.g., chives), the extensive use of phenylurea herbicides and dicarboximide fungicides results in high environmental risks owing to their hazardous nature of toxic residues. In addition, these pesticides are easy to be absorbed by crops and metabolized into 3,4-DCA or 3,5-DCA due to their excellent internal absorption features, and thus influence the quality and safety of agricultural products. It is urgent to develop an effective analytical method for 3,4-DCA and 3,5-DCA, which is significant to their residue monitoring in vegetable products (e.g., chives). Current detection methods for 3,4-DCA or 3,5-DCA include gas chromatography (GC), liquid chromatography (HPLC), gas chromatography-mass spectrometry (GC-MS), ultra-high performance liquid chromatography-tandem mass spectrometry (UPLC-MS/MS) and HPLC-MS/MS [25,26,27,28,29], but most methods were the single analytical method concerned with animal and soil samples for 3,4-DCA or 3,5 DCA, there were few reports on the simultaneous determination methods of 3,4-DCA and 3,5 DCA in plant samples. 

In this study, the chromatographic and mass-spectrometric conditions of HPLC-MS/MS were optimized using 3,4-DCA and 3,5-DCA standards. Meanwhile, the modified QuEChERS preparation technique suitable for 3,4-DCA and 3,5-DCA in chive products was developed by investigating the effects of different extractors, purifiers, and eluting agents on their recoveries based on the classical QuEChERS method. Compared with the original method, this method had the advantages of simplicity, rapidity, sensitivity, stability, and reliability. It was suitable for routine monitoring of 3,4-dichloroaniline and 3,5-dichloroaniline in vegetable products.

## 2. Materials and Methods

### 2.1. Chemical and Reagents

Standard substances of 3,4-DCA (98%) and 3,5-DCA (98%) were purchased from Shanghai Aladdin Biochemical Technology Co., Ltd. (Shanghai, China). HPLC grade acetonitrile (ACN) was purchased from Merck GMBH of Germany. HPLC grade formic acid (FA) was obtained from Shanghai Maclin Biochemical Technology Co., LTD (Shanghai, China). Other analytical grade reagents, such as primary secondary amine (PSA), graphite carbon black (GCB), C_18_ sorbent, florisil, ethyl acetate (EA), dichloromethane (DCM), acetone (ACE), methanol (MEOH), ammonia water (AA), benzene (BEN), toluene (MB) and acetic acid, were purchased from the Tianjin Kermel Chemical Reagent Co., Ltd. (Tianjin, China).

### 2.2. Stock and Standard Solutions

Standard stock solutions of DCA (100 mg/L) were prepared by dissolving 5 mg of 3,4-DCA and 3,5-DCA standards in 5 mL of AC and preserved in a refrigerator at 4 °C. The mixed standard stock solution (1 mg/L) preparation: 0.1 mL of 3,4-DCA and 3,5-DCA standards were mixed by dissolving them in 10 mL of AC. Solvent standard working curve preparation: 100 mg/L of 3,4-DCA and 3,5-DCA standard stock solution were diluted step by step with ACN into 0.001, 0.010, 0.050, 0.100, 0.500, and 1.000 mg/L of solvent standard mixture working solution, respectively.

### 2.3. Instruments and Analytical Conditions

The HPLC-MS/MS system comprised a 1290 Infinity II liquid chromatography LC system and an Agilent 6470A Triple Quadrupole mass spectrometer (Agilent Technology Co. LTD, Santa Clara, CA, USA). An Agilent Eclipse Plus C_18_ column (4.6 × 100 mm, 3.5 m particle size) was used as the chromatographic column. 0.1% FA aqueous solution (A)-AC (B) was used as the mobile phase, and the gradient elution procedure was as follows: 0.00–100 min, 70% B; 1.00–3.00 min, 70–90% B; 3.00–5.00 min, 90–70% B; 5.00–6.00 min, 70% B. The column temperature was set at 30; the flow rate was 0.5 mL/min, and the injection volume was 10 µL. Mobile phase A was 0.1% formic acid aqueous solution, and mobile phase B was acetonitrile with gradient elution. The elution procedure is shown in Table 1, and the retention time was 6 min.

A triple quadrupole tandem mass spectrometer equipped with electrospray ionization (ESI) was used to positively monitor 3,4-DCA and 3,5-DCA by multiple reaction monitoring (MRM). The sheath gas temperature and flow rate were 250 °C and 11.0 L/min; the nozzle voltage was 500 V; the capillary voltage was 3500 V; the atomization gas pressure was 45 psi; and the drying gas temperature and flow rate were 300 °C and 5 L/min, respectively. The above drying gas, atomization gas, collision gas, and sheath gas were all high-purity nitrogen. The MS/MS parameters of the two DCAs are shown in Appendix A.

### 2.4. Sample Preparation

An amount of 20 g of fresh chive samples was weighed and cut into small pieces of 1–2 cm, microwaved for 45 s, and ground with the appropriate amount of liquid nitrogen. Chive samples of 10.0 g were transferred to 50 mL polypropylene centrifuge tube, to which 20 mL of extractant (AC) was added and shaken for 5 min, 1 g of sodium chloride and 4 g of anhydrous magnesium sulfate were added, shaken for 5 min again, and then centrifuged at 4000 rpm/min for 5 min. Then, 1 mL supernatant and 0.25 mL MB were added to a 2 mL centrifuge tube with 150 mg of anhydrous magnesium sulfate and purification agent (10 mg GCB + 50 mg PSA), vortexed and shaken at 2000 rpm/min for 1 min, then centrifuged at 12,000 rpm/min for 3 min, followed by aspiration of 1 mL of supernatant using a disposable syringe. Prior to HPLC-MS/MS analysis, the supernatant was transferred to a 2 mL autosampler vial through a 0.22 μm organic filter and analyzed in triplicate.

### 2.5. Validation of the Method

Validation parameters were then evaluated according to FDA guidelines. To validate the analytical method of 3,4-DCA and 3,5-DCA in chive samples, blank aboveground and underground samples of chives were selected, and the verification parameters such as linearity, accuracy, repeatability, precision, limits of detection (*LOD*), and limits of quantification (*LOQ*) were assessed. The assessment of linearity was conducted by the matrix-matched calibration curve for 3,4-DCA and 3,5-DCA at the concentration of 0.001, 0.010, 0.050, 0.100, 0.500, and 1.000 mg/L in matrix blank extraction and ACN. The accuracy was estimated using the recovery (70–120%) tests by spiking the blank (DCAs-free) chive samples at four additional levels of 0.001, 0.010, 0.100, and 1.000 mg/kg. The relative standard deviation (*RSD*) (lower than 20%) of six replicates for each spiked level of 3,4-DCA and 3,5-DCA was used to assess the repeatability and precision of the analysis method. The concentration based on the signal-to-noise ratio of 3 was used as sensitivity (*LOD*), and the concentration based on the signal-to-noise ratio of 10 was used as *LOQ*. The matrix effect (*ME*) was calculated using the following formula:(1)$$ME=\frac{k_{m}-k_{s}}{k_{s}}\times 100\%$$
where, *k_m_* is the slope of the matrix-matching standard curve, *k_s_* is the slope of the pure solvent standard curve.

### 2.6. Statistical Analysis

All experiments were performed in at least triplicate, and data are expressed as mean ± relative standard deviation (% *RSD*). The results were statistically assessed by analysis of variance with a 5% significance level (*p* < 0.05), using the statistical software SPSS 20.0 to analyze the significance of the data.

## 3. Results and Discussion

### 3.1. Optimization of Mass Spectrum Conditions

The mass conditions for 3,4-DCA and 3,5-DCA were optimized from the following three stages, respectively.

First, the parent ions were determined: the scan mode of the mass spectrum was set to full scan mode (MS2 Scan), with a molecular weight scan range of 10–300 in positive ion mode and a fragmentation voltage of 135 V. As shown in Figure 1, the full-scan chromatogram of 3,4-DCA had four prominent chromatographic peaks, with retention time (RT) was 1.528, 2.268, 3.367 and 4.553 min, respectively. Moreover, their corresponding first-order mass spectrograms are shown as C, E, G, and I in Figure 1, respectively. The full-scan chromatogram of 3,5-DCA also had four prominent chromatographic peaks with retention times of 1.520, 1.775, 2.268, and 3.860 min, respectively, and their corresponding primary mass spectra are shown as D, F, H, and J in Figure 1, respectively. By analyzing the mass spectra of all chromatographic peaks, it can be seen that the mass-to-charge ratio of the higher response ions in the third chromatographic peak (RT = 3.367 min) of 3,4-DCA was 163.03 *m/z* and that in the third chromatographic peak (RT = 3.860 min) of 3,5-DCA was 163.03 *m/z*, which is the same as that of the higher response ions. Therefore, it can be preliminarily determined that these two chromatograms were the chromatographic peaks of 3,4-DCA and 3,5-DCA, respectively, and 163.03 *m/z* was its [M + H]^+^ ion peak. Therefore, 163.03 *m/z* was selected as the parent ion of 3,4-DCA and 3,5-DCA.

Secondly, for the parent ion fragmentation voltage optimization: The mass spectrometry scan acquisition mode was set to selective ion monitoring mode (MS2 SIM), and different fragmentation voltages (85, 90, 95, 100, 105, 110, 115, 120, 125, and 130 V) were applied to the selected parent ions (163.03 *m/z*). The optimal fragmentation voltage was determined by observing the response of parent ions at different fragmentation voltages. As shown in Figure 2, the parent ion abundance was highest for 3,4-DCA and 3,5-DCA when the fragmentation voltage was 115 V. Therefore, the fragmentation voltage was set to 115 V for 3,4-DCA and 3,5-DCA.

Thirdly, for the selection of daughter ions: The mass spectrometry scan acquisition mode was set to the product ion scan mode, and the product ion scan was performed at different collision energies (1, 20, 40, 60, and 80) for the parent ion (163.03 *m/z*) at the fragmentation voltage eV screened above, with a molecular weight scan range of 10–162. The positive ion mode was adopted, and the results are shown in Figure 3. The results show that under different collision energies, the high abundance of the daughter ions of 3,4-DCA and 3,5-DCA was 128.0 *m/z*, and the collision energy was 20 eV, followed by 75.1 *m/z*. At this time, the collision energy was 60 eV, thus the quantitative daughter ion of 3,4-DCA and 3,5-DCA was 128.0 *m/z*, with the collision energy of 20 eV, and the qualitative daughter ion was 75.1 *m/z*, with the collision energy of 60 eV.

### 3.2. Optimization of Chromatographic Conditions

The response intensities of isocratic and gradient elution were compared using two DCA solvent standards at 0.001 mg/L. As shown in Table 2, the peak area of 3,4-DCA at 0.001 mg/L increased by 29.05% under the gradient elution condition compared with that of isocratic elution, while the peak area of 3,5-DCA at 0.001 mg/L increased by 55.26% under this gradient elution condition compared with that of isocratic elution. This gradient condition was thus selected as the chromatographic condition for the detection of the two DCAs. According to the chromatographic and mass conditions screened above, 3,4-DCA and 3,5-DCA standards (0.001 mg/L) were detected. A, C, and D in Figure 4 represent the total ion flow, quantitative ion, and qualitative ion chromatogram of 3,4-DCA, respectively, and B, D, and F represent the total ion current chromatogram, quantitative ion chromatogram, and qualitative ion chromatogram of 3,5-DCA, respectively. The retention time was 3.352 min for 3,4-DCA and 3.851 min for 3,5-DCA. The chromatograms showed good peak shape and few spurious peaks, which can be used for the quantitative analysis of the two DCAs.

### 3.3. Optimization of Extractants in Chive Samples

Common extractants for vegetable pesticide residues are mainly organic solvents such as ACN, EA, DCM, ACE, and ME, among others. According to literature reports, an addition of a small number of organic acids (such as FA and acetic acid) or AA changing the pH of the extraction environment during the extraction process can enhance the recovery of some pesticides [29,30]. In this study, the addition of a small number of organic acids or AA was thus considered for extracting 3,4-DCA and 3,5-DCA from chive samples.

For 3,4-DCA, 1 mg/L standard solution was added to the blank sample of chives, the extraction effects of ACN, 1% FA-ACN, 1% AA-ACN, DCM, EA, and ACE were investigated, and the results are shown in Figure 5A. For the aboveground tissue samples of chives, the extraction recoveries of AC were the highest (91.7%), followed by 1% FA-ACN (89.9%), 1% AA-ACN (88.6%), EA (88.2%), ACE (82.3%), and DCM (79.8%). Thus, ACN was selected as the extraction agent for aboveground tissue samples of chives. For the underground tissue samples of chives, the extraction recoveries of ACN were 88.4%, followed by 1% FA-ACN (88.1%), 1% AA-ACN (86.8%), EA (82.7%), DCM (77.6%), and ACE (75.6%). Likewise, can was selected as the extraction agent for the underground tissue samples of chives. 

For 3,5-DCA, the extraction was also carried out with the same extractants and processes as those described above for 3,4-DCA. As indicated in Figure 5B, for the above ground tissue samples of chives, the extraction recoveries of 1% AA-ACN were 100.9%, followed by ACN (98.1%), 1% FA-ACN (97.0%) DCM (94.9%), EA (87.8%), and ACE (79.5%). Since there were no significant differences in the extraction recoveries between ACN and 1% AA-ACN, ACN was also used as the extraction agent for the chive aboveground tissue samples. For the chive underground tissue samples, the extraction recovery of ACN was the highest (98.8%), followed by 1% FA-ACN (90.2%), 1% AA-ACN (86.4%), DCM (84.5%), EA (79.6%), and ACE (76.7%). AC was thus selected as the extractant for the chive underground tissue samples.

### 3.4. Evaluation of the Purification Effects on Chive Samples 

Florisil, C_18_, PSA, and GCB are commonly used purification materials for the QuEChERS method of pesticides [31]. C_18_ can be used to remove lipids. PSA can eliminate organic acid pigments, sugars, and fatty acids in various vegetables and fruits. GCB can be used to remove pigments such as steroid chlorophyll [32]. To effectively remove the pigment in chive samples and ensure a satisfactory recovery, the effect of the amount of GCB and ME on the recovery was investigated. The method of the orthogonal test was thus adopted. The ratio of the eluting agent ACN-ME was set as 1:0, 1:1, 2:1, 3:1, 4:1, and 5:1 (*v*/*v*). The amount of GCB was set as 5, 10, 15, 20, and 25 mg, respectively. As shown in Figure 6, when ME was not added into the supernatant, the amount of GCB was between 5 and 25 mg, and the recovery of 3,4-DCA and 3,5-DCA was respectively 9.9–54.6% and 16.9–70.1%, indicating that GCB had a strong adsorption capacity for 3,4-DCA and 3,5-DCA. It can also be seen from Figure 6 that after adding a certain amount of MB, the recovery of 3,4-DCA and 3,5-DCA was significantly improved and increased by 9.5–66.9% and 5.6–63.3%, respectively. However, according to the color depth of the sample bottle shown in Figure A1 of Appendix B, the addition of ME also reduced the depigmentation ability of GCB to a certain extent. The ACN-MB ratio of 4:1 (*v*/*v*) was selected eventually. When the amount of GCB was 10 mg, the recovery of 3,4-DCA and 3,5-DCA at the ratio of ACN-MB 4:1 (*v*/*v*) was 84.2% and 94.2%, which also had good depigmentation ability. Meanwhile, the dosage of C_18_, PSA, and florisil was 50 mg in this experiment.

Further analysis of the effects of BEN and MB on the recovery of two DCAs shows that the difference between BEN and MB was less than 5%, indicating that there was little difference in the elution effect of two DCAs between BE and MB. As shown in Figure 7, however, when BEN was added, the peak of 3,5-DCA in the chromatogram had an evident trailing phenomenon. Therefore, MB was finally selected, which was, together with ACN, used as the eluting agent in this study.

Six combinations were used to purify two DCAs in aboveground tissue samples of chives at the spiked level of 1 mg/kg, and the results are shown in Figure 8A. For 3,4-DCA, the recovery after purification with GCB + florisil was highest (86.5%), followed by GCB + PSA (86.2%), GCB + PSA + florisil (84.8%), GCB + C_18_ (84.1%), GCB + C_18_ + florisil (83.3%), and GCB + PSA + C_18_ (81.1%). Since there were significant differences among three combinations of GCB + PSA, GCB + PSA + florisil, and GCB + C_18_, and GCB + PSA was the most commonly used purification combination in vegetables, 10 mg GCB + 50 mg PSA was selected as the purifying agent for 3,4-DCA in the aboveground tissue samples of chives. For 3,5-DCA, the recovery with GCB + PSA (94.8%) was the highest, followed by GCB + florisil (93.9%), GCB + C_18_ (92.48%), GCB + PSA + florisil (87.5%), GCB + C_18_ + PSA (82.9%), and GCB + C_18_ + florisil (80.6%). Therefore, 10 mg GCB + 50 mg + PSA was also selected as the purifying agent for 3,5-DCA in the aboveground tissue samples of chives. Similarly, the purification of two DCAs from chive underground tissue samples at the spiked level of 1 mg/k was carried out using the same six combinations, and the results exhibit that the recovery of 3,4-DCA and 3,5-DCA purified by the combination of 10 mg GCB + 50 mg PSA6 was highest (Figure 8B), which was also selected as the purifier of 3,4-DCA and 3,5-DCA in chive underground tissue samples.

### 3.5. Method Validation

#### 3.5.1. Linearity, LOD, and LOQ

Based on the optimized chromatographic and mass-spectrometric condition and sample preparation technique mentioned above, the linearity was assessed by the matrix-matched calibration curve. As exhibited in Table 3, both 3,4-DCA and 3,5-DCA showed a good linear relationship in the range of 0.001–1.000 mg/L (*R*^2^ > 0.996). Values of LOD and LOQ of 3,4-DCA in the aboveground tissue of chives were 0.6 and 2.0 μg/kg, respectively. Values of LOD and LOQ of 3,5-DCA in aboveground tissue of chives were 1.0 and 3.0 μg/kg, respectively. ME is not negligible when HPLC-MS/MS is used for the detection of pesticide residues [33]. It can be seen in Table 3, except that the matrix of underground tissue of chives showed weak matrix enhancement for 3,5-DCA; the matrix of the rest was matrix inhibition. The matrix inhibition/increase of all the substrates for the two DCAs was not more than 20%, which further indicates that the sample preparation technique mentioned above had better extraction and purification effect.

#### 3.5.2. Accuracy and Precision

Four spiked levels (0.001, 0.010, 0.100, and 1.000 mg/kg) were set for all samples, with six replicates for each level. According to the SANTE [34] and CODEX [35] guidelines, the recovery is used to evaluate the accuracy of the method. If the recovery is within 70–120%, it indicates that the accuracy of the present method is acceptable. RSD is used to evaluate the precision of the method, and if RSD is less than 20%, it means that the precision of the method is acceptable. As shown in Table 4, the fortified recoveries of 3,4-DCA in aboveground and underground tissues of chives were 75.3–85.4% and 78.5–86.0%, with RSD of 2.1–8.5% and 2.7–6.7%, respectively. Meanwhile, the fortified recoveries of 3,5-DCA in aboveground and underground tissues of chives were 78.2–98.1% and 79.6–93.4%, with RSD of 2.1–9.4% and 1.4–11.9%, respectively. The fortified recoveries of all spiked levels were in the range of from 70% to 120%, and corresponding RSDs were less than 20%, indicating that the method had satisfactory accuracy, repeatability, and precision, as well as could meet the requirements of the residue analysis of the two DCAs in chive products.

In the analysis process, the key to accurately quantifying the target analyte is the sample pretreatment process. QuEChERS is a quick, simple, cheap, effective, robust, and safe sample pretreatment method developed by Anastassiades et al. in 2003 [36]. This method was first proposed for the detection of pesticide residues in fruits and vegetables and has been cited and optimized by more researchers. As shown in Appendix A, the optimization of the classical QuEChERS method in this study was much simpler in steps. Compared to the classical QuEChERS method, cumbersome steps such as concentration and re-solubilization were eliminated, uses fewer reagents, and takes less time than the QuEChERS method in GB 23200.113-2018, and was more suitable for the simultaneous and rapid detection of 3,4-DCA and 3,5-DCA in chives. 

### 3.6. Application of Method for the Analysis of Real Samples

The developed method was also used to analyze sixty vegetable samples from vegetable production bases. The residues of 3,4-DCA and 3,5-DCA in the vegetable samples analyzed in this study are shown in Table 5. According to the National food safety standard—Maximum residue limits (MRLs) for pesticides in food (GB 2763-2021), the residues of 3,4-DCA and 3,5-DCA, as well as their parent compounds diuron, propanil, iprodione, and procymidone in vegetables met the requirements of pesticide MRLs.

## 4. Conclusions

In conclusion, a validated simultaneous analysis method was developed to rapidly determine 3,4-DCA and 3,5-DCA residues in chive products based on the optimized HPLC-MS/MS condition and modified QuEChERS method. The modified QuEChERS preparation method effectively used ACN extraction, GCB+PSA purification, and then ACN+ME elution, removing the interference and reducing the matrix effect of 3,4-DCA and 3,5-DCA in chive products. The analytical method showed satisfactory accuracy (recoveries of 70–120%) and precision (RSD < 20%). In addition, a quick and efficient QuEChERS method was developed and substituted the classical QuEChERS method consisting of extraction steps. Moreover, this method was simple, rapid, sensitive, stable, and reliable for the routine monitoring of 3,4-DCA and 3,5-DCA in vegetable products. The method was successfully applied to real vegetable samples.

## Figures and Tables

**Figure 1 foods-12-02875-f001:**
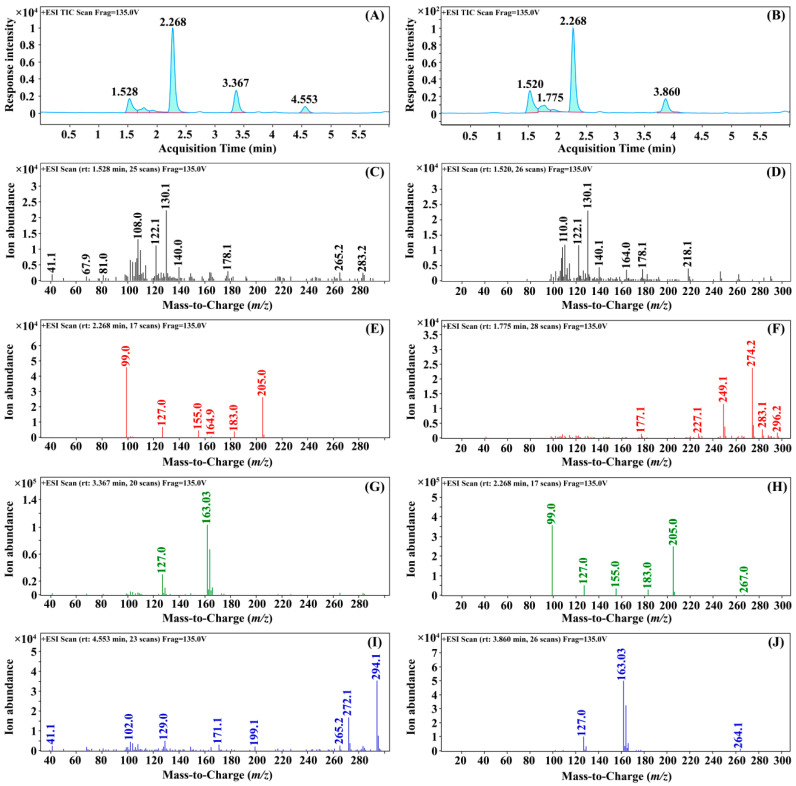
The full-scan chromatogram and mass spectra of 3,4-DCA and 3,5-DCA. Note: (**A**,**B**) is the total ion current chromatogram of 3,4-DCA and 3,5-DCA, respectively. (**C**,**E**,**G**,**I**) is the primary mass spectra of 3,4-DCA at 1.528, 2.268, 3.367, and 4.553 min, respectively. (**D**,**F**,**H**,**J**) is the primary mass spectra of 3,5-DCA at 1.520, 1.775, 2.268, and 3.860 min, respectively.

**Figure 2 foods-12-02875-f002:**
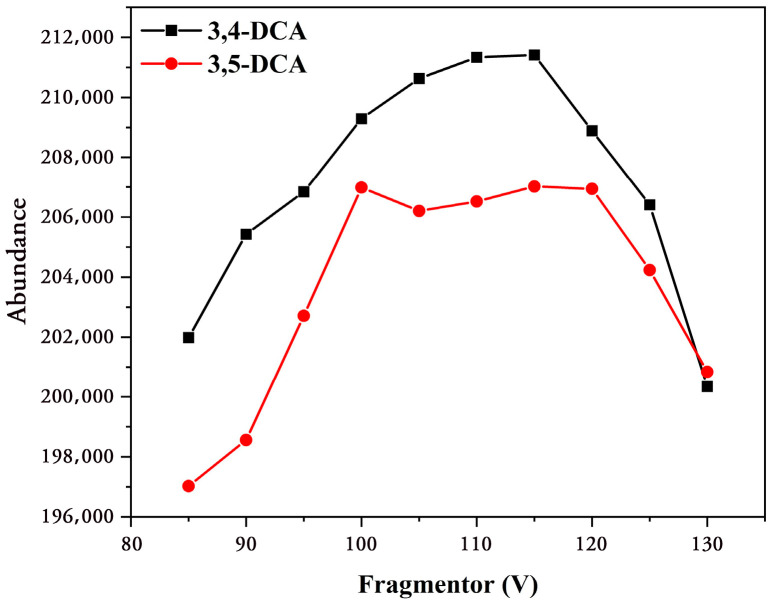
The effect of different cone voltages on the abundances of 3,4-DCA and 3,5-DCA parent ions.

**Figure 3 foods-12-02875-f003:**
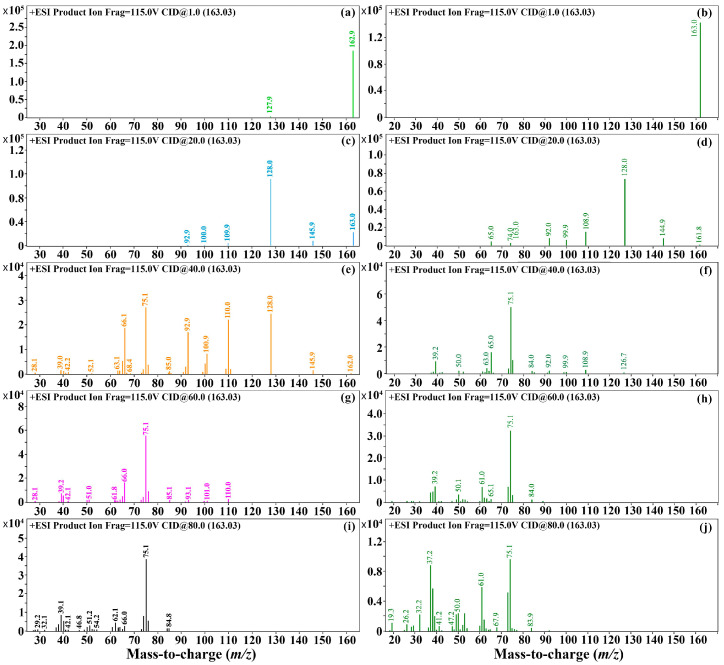
The secondary mass spectra of 3,4-DCA and 3,5-DCA at different collision energies. Note: (**a**,**c**,**e**,**g**,**i**) is the second-order mass spectra of 3,4-DCA at different collision energies of 1, 20, 40, 60, and 80 eV, respectively. (**b**,**d**,**f**,**h**,**j**) is the second-order mass spectra of 3,5-DCA at different collision energies of 1, 20, 40, 60, and 80 eV, respectively.

**Figure 4 foods-12-02875-f004:**
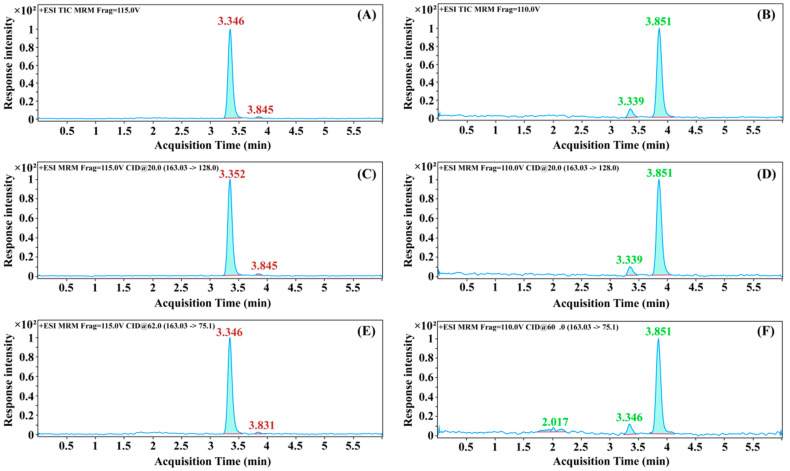
The total ion current chromatogram, quantitative ion chromatogram, and qualitative ion chromatogram of 3,4-DCA and 3,5-DCA. Note: (**A**,**C**,**D**) is the total ion current chromatogram, quantitative ion chromatogram, and qualitative ion chromatogram of 3,4-DCA, respectively. (**B**,**D**,**F**) is the total ion current chromatogram, quantitative ion chromatogram, and qualitative ion chromatogram of 3,5-DCA, respectively.

**Figure 5 foods-12-02875-f005:**
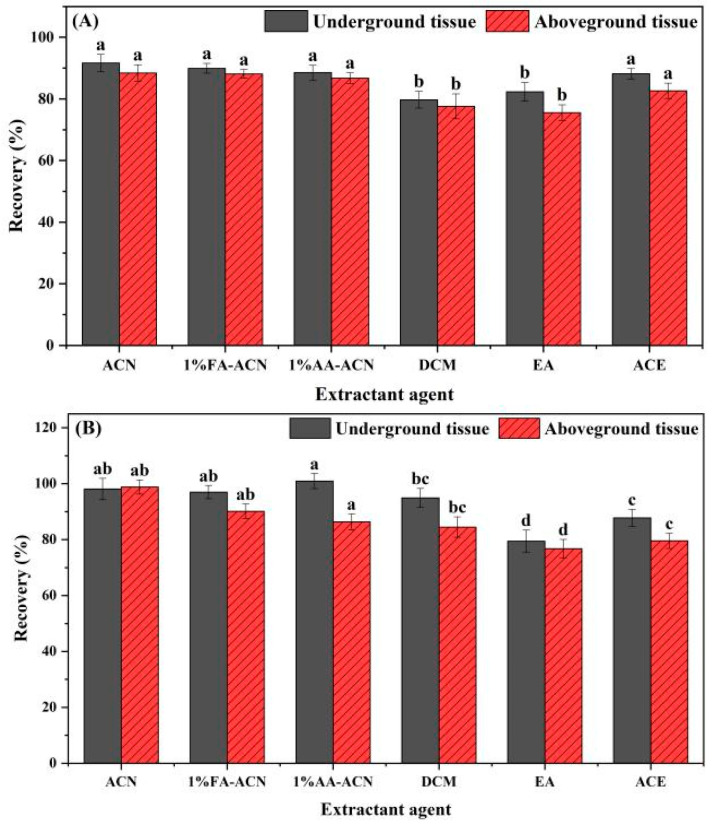
The extraction recovery of 3,4-DCA (**A**) and 3,5-DCA (**B**) from chive samples with different extractants. Note: Each experiment was repeated six times in each group. The column is marked with different letters indicating significant differences (*LSD*, *p* < 0.05).

**Figure 6 foods-12-02875-f006:**
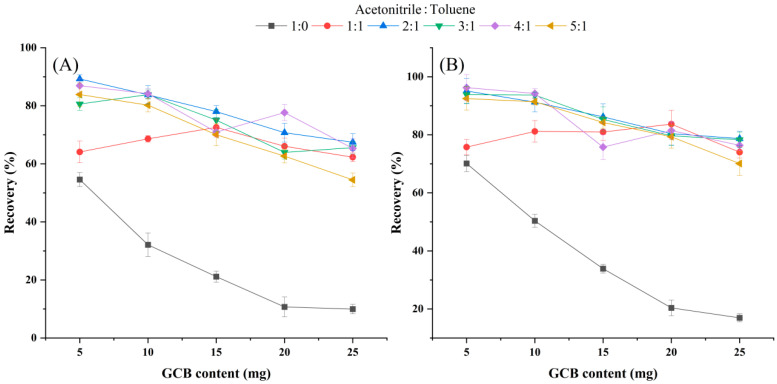
The effect of the different GCB and MB contents on the recovery of 3,4-DCA (**A**) and 3,5-DCA (**B**). Note: Each experiment was repeated six times in each group.

**Figure 7 foods-12-02875-f007:**
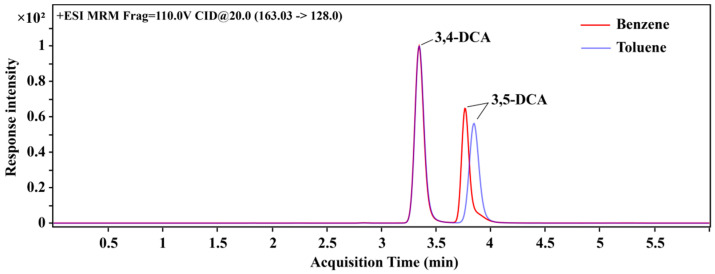
The chromatogram of 3,4-DCA and 3,5-DCA with the addition of MB and BEN.

**Figure 8 foods-12-02875-f008:**
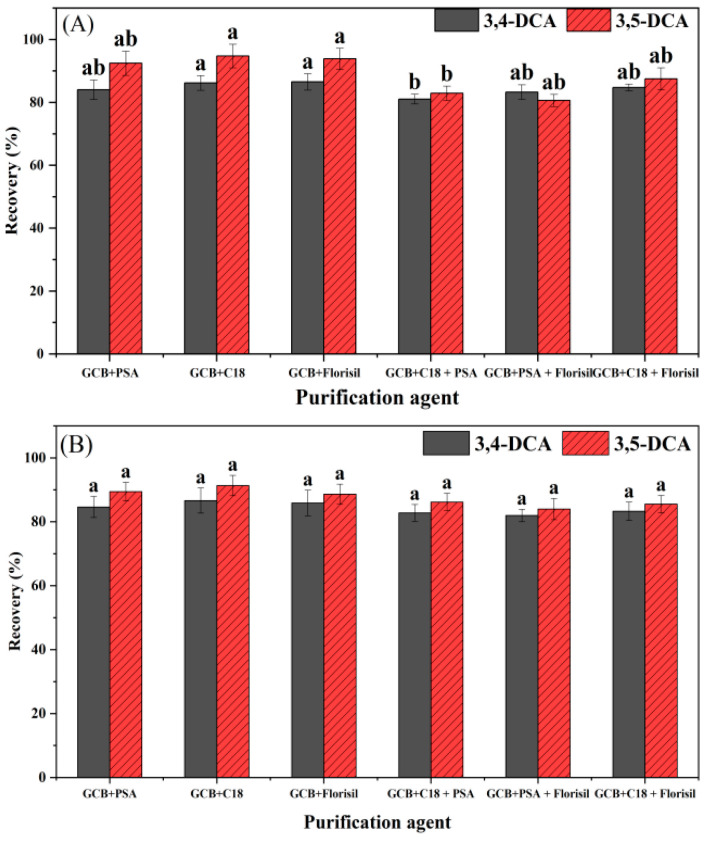
The effect of different purification agents on recoveries of 3,4-DCA and 3,5-DCA in aboveground (**A**) and underground (**B**) tissue samples of chives. Note: Each experiment was repeated six times in each group. The column is marked with different letters indicating significant differences (LSD, *p* < 0.05).

**Table 1 foods-12-02875-t001:** Gradient elution conditions for ultra-high performance liquid chromatography.

Retention Time/min	Flow Rate/(mL/min)	Mobile Phase A/%	Mobile Phase B/%
0	0.5	30	70
1	0.5	30	70
3	0.5	10	90
5	0.5	30	70
6	0.5	30	70

**Table 2 foods-12-02875-t002:** The effect of isocratic elution and gradient elution on the peak area of 3,4-DCA and 3,5-DCA.

Type of Elution	Compound	Peak Area 1	Peak Area 2	Peak Area 3	Mean Peak Area	Response Improvement Rate (%)
Isocratic elution	3,4-DCA	605.95	666.08	585.39	619.14	—
	3,5-DCA	298.19	286.52	307.86	297.52	—
Gradient elute	3,4-DCA	742.75	744.61	779.67	755.68	22.05
	3,5-DCA	458.88	465.01	461.95	461.95	55.26

**Table 3 foods-12-02875-t003:** The linear equation, *R*^2^, LOD, LOQ, and ME of 3,4-DCA and 3,5-DCA in different substrates.

Compound	Matrix	Linear Equation	*R* ^2^	*LOD* (μg/kg)	*LOQ* (μg/kg)	*ME* (%)
3,4-DCA	Aboveground tissue	y=612507.83x+10221.63	0.9972	0.6	2.0	−9.0
	Underground tissue	y=655299.87x−665.51	0.9999			−2.6
	Solvent	Y=673040.70x−1117.69	0.9999 *	—	—	—
3,5-DCA	Aboveground tissue	y=320574.71x+5473.42	0.9969	1.0	3.0	−4.4
	Underground tissue	y=342761.98x+782.62	0.9999			2.3
	Solvent	y=335172.19x+151.05	0.9999 *	—	—	—

Note: * indicates that *R*^2^ is greater than 0.9999.

**Table 4 foods-12-02875-t004:** The recovery and RSD of 3,4-DCA and 3,5-DCA in chive samples at different spiked levels (n = 6).

Compound	Matrix	Addition Level (mg/kg)	Average Recovery (%)	*RSD* (%)
3,4-DCA	Aboveground tissue	0.001	75.3	8.5
		0.010	85.4	4.3
		0.100	80.1	2.6
		1.000	84.4	2.1
	Underground tissue	0.001	78.5	6.7
		0.010	77.3	3.8
		0.100	81.2	3.2
		1.000	86.0	2.7
3,5-DCA	Aboveground tissue	0.001	78.2	9.4
		0.010	98.1	7.5
		0.100	87.1	5.1
		1.000	94.3	2.1
	Underground tissue	0.001	79.6	10.6
		0.010	82.0	11.9
		0.100	87.2	2.9
		1.000	93.4	1.4

**Table 5 foods-12-02875-t005:** The concentration 3,4-DCA and 3,5-DCA in vegetable samples.

Compound	Parent Compounds	Vegetable	Average (mg/kg)	Median (mg/kg)	Range (mg/kg)
3,4-DCA	Diuron	Chive	0.076	0.082	0.01–0.176
		Carrot	0.016	0.024	0.002–0.053
		Onions	—	—	—
		Leek	—	—	—
		Parsley	—	—	—
	Propanil	Spinach	—	—	—
		Tomato	—	—	—
		Cucumber	—	—	—
		Potato	—	—	—
		Pepper	—	—	—
3,5-DCA	Iprodione	Chive	0.083	0.067	0.01–0.2
		Leek	0.273	0.312	0.02–1.07
		Tomato	0.084	0.067	0.01–0.154
		Onion	—	—	—
		Cucumber	—	—	—
	Procymidone	Potato	—	—	—
		Pepper	—	—	—
		Parsley	—	—	—
		Spinach	—	—	—
		Carrot	—	—	—

## Data Availability

Data is contained within the article or Appendix A.

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
