# Peer review of "Development of Simultaneous Determination Method of Pesticide High Toxic Metabolite 3,4-Dichloroaniline and 3,5 Dichloroaniline in Chives Using HPLC-MS/MS"

_foods, 2023, doi:10.3390/foods12152875_

Round 1

Reviewer 1 Report (New Reviewer)

Comments:

Comment-1: The abstract fails to provide sufficient background information regarding the significance of 3,4-dichloroaniline (3,4-DCA) and 3,5-dichloroaniline (3,5-DCA) residues in chive products. It would be better to include 2-3 sentences explaining why these metabolites are of concern, their potential health risks, and their prevalence in the environment or food products. Authors should add essential details about the methodology employed in the study (specific HPLC-MS/MS conditions & parameters etc.). It would be more appropriate if authors add conclusion or key findings of the study in abstract section.

Comment-2: The introduction lacks a clear structure and organization. Authors should provide a concise overview of the background, research gap, and objectives of the study. The information presented seems to be scattered and lacks a logical flow. The authors should clarify why the existing methods are insufficient or inadequate for analyzing these compounds (3,4-DCA, 3,5-DCA) specifically in chives.

Comment-3: Authors mentioned the use of a modified QuEChERS preparation technique in the manuscript but it lacks specific details on the modifications made. It is important to describe the modifications in detail and explain why they were necessary or beneficial for the analysis of 3,4-DCA and 3,5-DCA residues in chive samples.  The description of sample preparation is insufficient. The manuscript mentions the use of 20 g of chive samples but does not specify the nature of the samples (e.g., fresh, dried), the number of replicates, or any specific guidelines followed during the sample preparation process. These details are crucial for ensuring reproducibility and understanding the validity of the results.

Comment-4: The manuscript contains several grammatical and formatting issues, such as inconsistent use of spacing, inconsistent use of punctuation, and incomplete sentences.

Comment-5: Authors should provide a comparison of the developed method with other existing methods for the analysis of 3,4-DCA and 3,5-DCA. Including a discussion on the advantages and limitations of the developed method compared to other methods would provide valuable insights and help position the study in the existing scientific literature.

Comment-6: The manuscript does not mention any statistical analysis performed on the data. It is important to include statistical tests or methods used to evaluate the significance of the results and the reliability of the method.

Comment-7: Limitations of the study should be discussed along with future perspective for further research.

Comment-8: Concise summary of the key findings of the study should be provided regarding the detection and quantification of 3,4-DCA and 3,5-DCA residues in chive products and their significance in terms of food safety or regulatory compliance.

Comment-9: It would be better to incorporate the abbreviations list used in the manuscript.

The manuscript contains several grammatical and formatting issues. Minor English editing is required

Author Response

Reviewer 2 Report (Previous Reviewer 1)

The reviewed manuscript is a resubmission to the manuscript foods-2257915. The authors addressed the concerns raised by the Reviewer. However, the English language still needs revision since many grammatical errors have to be corrected prior to the publication of the manuscript.

The English language still needs revision since many grammatical errors have to be corrected prior to the publication of the manuscript.

Author Response

Reviewer 3 Report (New Reviewer)

The manuscript titled "Development of Simultaneous Determination Method of Pesticide High Toxic Metabolite 3,4-Dichloroaniline and 3,5 Dichloroaniline in chives Using HPLC-MS/MS" by Dong et al. aims to optimize the chromatographic and mass-spectrometric conditions of HPLC-MS/MS using 3,4-DCA and 3,5-DCA standards and to develop a modified QuEChERS preparation technique suitable for 3,4-DCA and 3,5-DCA in chive products by investigating the effects of different extractors, purifiers and eluting agents on their recoveries based on the classical QuEChER.

Some comments should be addressed for improving this manuscript:

- In the first and second lines of abstract, there are a grammar mistake that needs to be corrected.

- Table 1 and 2 can be transferred into supplementary data instead and make the required changes in the sequence of tables depending on that.

- under this subtitle "3.2. Optimization of chromatographic conditions", please revise the sentence starting with "As exhibited in Figure 4, A, C and ......".

- Also figure 9 can be transferred into supplementary data.

- Please revise the whole manuscript for improving the language.

Minor revision of English language is required.

Author Response

Reviewer 4 Report (New Reviewer)

This article is dealing with a method development and validation for metabolites' 3,4-dichloroaniline and 3,5-dichloroaniline simultaneous determination in chive using liquid chromatography-tandem mass spectrometry and modified QuEChERS sample preparation technique. Thus, the subject of the article corresponds with the aims of Special Issue "Rapid Analytical, Removal and Transformation of Chemical Residues in Foods". Research background seems to be sufficiently presented and the manuscripts itself well prepared. Used methods, i.e. instrumental and sample prep method development, validation procedure, chosen method performance characteristics and acceptance criteria, are adequately described and the research results adequately presented in figures and tables. Cited references are relevant although not entirely current and provided conclusion suits the manuscript content and research findings. Therefore, I have no major objections for this manuscript to be published, however I leave a few remarks for authors to address:

Abstract In this study, a validation high-performance liquid chromatography-tandem mass spectrometry (HPLC-MS/MS) method was developed… Validation is an excess?

Page 3 − …urgent to develop a effective analytical method… An effective?

3.6. Application of method – The residues of 3,4-DCA and 3, 5-DCA in all samples were below the maximum residue limits of parent compounds. Could indicate which parent compounds and where the MRLs are set?

Author Response

Reviewer 5 Report (New Reviewer)

Dear author , 

the manuscript addressed a good idea and I have some little comments for you:

- the part of conclusion needs some details to be more informative 

- the references needs to be up to date , its seems old

regards 

Author Response

This manuscript is a resubmission of an earlier submission. The following is a list of the peer review reports and author responses from that submission.

Round 1

Reviewer 1 Report

In this manuscript, a UPLC-MS/MS method was developed for the determination of  3,4-Dichloroaniline and 3,5 Dichloroaniline in chive samples. The herein study is well-organized and it can be characterized as interesting and novel. The literature review is comprehensive. However, there are some issues and considerations that must be corrected.

-The English language needs revision since there are many grammatical errors that have to be corrected. For example “a validation UPLC-MS/MS analytical method”, “chromatographic and mass spectrometry conditions”, “exhibits that acetonitrile”. Please check the whole manuscript for such errors.

-Introduction: Please make a separate paragraph to mention what was done in this study (i.e., in this study, the chromatographic…….was established.”

-The UPLC-MS/MS instrument must be mentioned in the study.

-Please replace ~ with – throughout text.

-“The injection volume was 1 L”, please correct the unit.

-Please add the number of replicates for the optimization experiments in the text and in the figures where error bars are used.

-Please replace the abbreviations with more widely used ones. For example, ACN for acetonitrile, MeOH for methanol etc.

-The description of Figure 6 must be shortened. “The further analysis….in this study” can be a separate paragraph of the main text.

-Did the authors use the method for real chive sample analysis? A respective section must be included demonstrating the results for real samples.

Reviewer 2 Report

In this study, a novel UPLC-MS/MS method was developed and validated for simultaneous determination of high toxic metabolite 3,4-Dichloroaniline and 3,5 Dichloroaniline in chives. Both 3, 4-dichloroaniline (3, 4-DCA) and 3, 5-dichloroaniline (3, 5-DCA) is primary metabolite deriving from the breakdown of phenylurea herbicides and dicarboximide fungicides, whose residue in vegetables has a heightened concern and toxic health risks in humans. Authors claim novelty as this is first simultaneous analysis of subject analytes in chives. Overall the manuscript is well written and following concerns must be address by authors before its consideration in foods.

1.  In introduction, authors mentioned that only few methods reported for simultaneous anlaysis…..Therefore, advantage of this method over reported method need to be discussed here.

2.       Section 2.3, The reported method is UPLC-MS/MS, but the column used here is (4.6x 100 mm, 3.5 m particle size) which seem to be normal C18 COLUMN. How this column based separation can be considered as UPLC separation. Authors need to mention the details of instrument (UPLC and TQD detector) including manufacturer name and address.

3.       How the injection volume of 1 L can be injected to UPLC sytem?

4.       On what basis different cone voltage and collision energy was selected for MS and MS/MS transition optimization?

5.       Its seem that no internal standard used for this method? Then how the expected loss of samples during extraction was compensated?

6.       The cone voltage and collision energy for both meatabolites were same? Then how these analyte was separated from each other?

7.       Figure 4. Authors need to provide the blank and LOQ/LOD chromatograms as represented chromatogram.

8

Reviewer 3 Report

The topic presented in this manuscript is quite interesting, but the introduction should be improved, calculations are sometimes wrong, one table is missing and another is unclear. Furthermore, the English is very difficult to understand/incomprehensible and it needs a deep revision. 

Round 2

Reviewer 2 Report

Authors have addressed most of the comments however I am still not satisfied with response of point no 3.  Authors used column (4.6x 100 mm, 3.5 m particle size) which seem to be normal C18 COLUMN therefore the method can be considerd as HPLC-MS/MS method not uplc-ms/ms method. 

Author should changed the UPLC-MS/MS by HPLC/MS/MS throughout the manuscript including title.

Reviewer 3 Report

The language is still entirely inappropriate. Extensive editing of the English language and style is still required before publication. I do not suggest the publication of the manuscript in this form. 
